# Comparative Analysis of Lead Removal from Liquid Copper by ICF and CCF Refining Technologies

**DOI:** 10.3390/ma15197024

**Published:** 2022-10-10

**Authors:** Leszek Blacha, Albert Smalcerz, Bartosz Wecki, Jerzy Labaj, Debela Geneti Desisa, Maciej Jodkowski

**Affiliations:** 1Department of Metallurgy and Recycling, Faculty of Materials Science, Silesian University of Technology, Krasinskiego 8, 40-019 Katowice, Poland; 2Department of Industrial Informatics, Faculty of Materials Science, Silesian University of Technology, Krasinskiego 8, 40-019 Katowice, Poland; 3Department of Testing and Certification “ZETOM”, Ks. Herberta Bednorza 17, 40-384 Katowice, Poland; 4Department of Industrial Informatics, Faculty of Materials Science, Joint Doctorate School, Silesian University of Technology, Krasinskiego 8, 40-019 Katowice, Poland

**Keywords:** induction melting, vacuum refining, meniscus, interface, non-ferrous metals, purification of copper, mass transport coefficient

## Abstract

Innovative technologies require the use of materials that meet increasingly high requirements; one such requirement is the purity of metals. In the case of copper, this translates into a parameter related to electrical conductivity. Traditional metal refining technologies have some limitations that can be eliminated through the use of modern melting aggregates. Such solutions include vacuum induction furnaces, comprising an induction furnace with a cold crucible. As part of this work, the possibilities of refining copper and lead alloys were investigated. In addition, the research was carried out with the use of two induction vacuum aggregates, allowing us to compare their effectiveness. The tests were carried out in a pressure range of 10–1000 Pa and at temperatures of 1273–1473 K. The results obtained made it possible to determine the mass transport coefficient of lead from an alloy with copper, and to determine the share of resistance in individual stages of the process. For experiments conducted inside an induction crucible furnace, lowering the working pressure inside the furnace chamber from 1000 to 10 Pa while increasing the temperature from 1323 to 1473 K was accompanied by a drop in the lead concentration inside the alloy of 69 to 96%, compared to its initial mass. For experiments conducted inside a cold crucible furnace, approximate values of lead removal appeared for lower temperatures (1273 to 1323 K), confirming that the analyzed process happens faster in this aggregate.

## 1. Introduction

In recent years, demand for metallic materials with special properties has increased. It has forced the development of technologies that allow for the production of metals and alloys of high purity. At the same time, the significance of vacuum smelting technology has also grown. Vacuum metal refinement technologies are based on the phenomena of the evaporation of metal bath components characterized by high vapor pressure, and the separation of gasses contained within the metal bath. The process of evaporation is complex, due to its heterogenic character. The temperature and pressure within the molten aggregate, its fluid dynamics, and the alloy chemical structure can strongly impact the rate of the analyzed process. Determining which of these properties has the most significant impact on the kinetics of the evaporation process and determining the character of the control of this process require multiple experiments to be conducted.

Vacuum induction melting (VIM) is a commonly used melting technique which involves melting an alloy in a vacuum or in an inert atmosphere by electromagnetic induction using coils. The main source of power during the process is the alternative magnetic field formed by an alternate current passing through an induction coil. Compared to other techniques, the use of VIM technology for metals alloys in refractory crucibles is much less energy intensive [1,2,3]. In addition, VIM allows for the fast homogenization of the melt by electromagnetic stirring, and is less expensive than alternative melting methods. Despite its advantages, VIM has not been frequently applied to all alloys, largely due to the lack of stable conventional refractory crucibles for reactive metals. Cold crucible melting, which suspends the molten metal using an electromagnetic force [4,5,6], has recently received considerable attention for the melting of reactive and refractory metals. Induction has a unique feature, in that the liquid metal is kept in a solid skull of the same metal without contacting the crucible, and is stirred by an electromagnetic force [7,8,9,10].

In the work presented here, we show the examination results of the rate with which lead was removed from copper alloys molten within a vacuum induction crucible furnace (ICF) and a cold crucible furnace (CCF). This specific type of examination was conducted due to the fact that the contamination examined strongly lowers the electric conductivity of copper. The devices used were chosen because induction smelting causes a significant increase in the surface of the metal bath, which can intensify the analyzed process of refining copper from lead.

## 2. State of the Art

The first literature references related to the research of copper or copper alloy vacuum refining were published in the 1960s and in the first half of the 1970s [11,12,13,14]. The results presented in these works were limited, presenting only the data that showed the change in the concentration of the removed additives during the smelting process. Among the data missing from these papers is the information regarding the chemical composition of the alloys examined, the length of the process, and the type of smelting aggregate. The first data regarding the kinetics of lead removal from liquid copper in the process of refining in a vacuum were published by Ohno [15]. He examined the rate of the removal of Pb, Ag, and Bi from synthetic copper alloys, focusing on the influence of the oxygen and sulfur concentrations within the alloy and the rate of bath mixing in this process. According to this author, the process of lead evaporation was controlled by mass transport in the gas phase due to evaporation. In 1977 [16], the same author presented the results of examinations aiming to show the influence of metal bath mixing on the rate of the removal of these same copper contaminants. The research was conducted using a vacuum aggregate, inside which the metal sample was heated with an electric resistive furnace. The metal bath was mixed with a molybdenum mixer. The results showed that increasing the mixing rate does not strongly corelate with the removal of the analyzed contaminants from copper.

Further papers published by Ozberk and Guthrie [17,18] contained results of examinations of the removal of copper, antimony, bismuth, and arsenic from liquid copper, conducted using a vacuum induction aggregate. Based on their experimental results, these authors concluded that the process of lead and bismuth evaporation is primarily controlled by mass transport in the gas phase. However, lowering the pressure inside the system to around 8 Pa and increasing the bath temperature above 1623 K causes changes in the process control. Under these conditions, the evaporation process for the mentioned contaminants is controlled by mass transport both in the gas phase and in the liquid phase. The results of copper vacuum refinement, also from lead-containing samples, have been presented in previous works [19,20,21,22]. In papers [19,20,21], the results of the vacuum refinement of a Cu-Pb alloy and blister copper in the temperature range of 1373–1523 K and within a wide range of pressures (8–1333 Pa) are discussed. In these works, it is shown that for pressures of below 80 Pa, the analyzed evaporation process of lead is determined by gas transport within the gas phase. As the pressure is reduced, the resistance of mass transport within the gas phase is lowered; at the same time, the relative resistance of mass transport within the liquid phase increases.

From these works, only [21] addresses the removal of lead from liquid Cu in the process of vacuum refinement in vacuum induction furnaces. The values of the lead mass coefficient inside the liquid phase are linked to the value of the near-surface rate of liquid copper stirred using induction. In [21,22], the phenomenon of a liquid copper meniscus appearing was taken into account regarding the rate of lead evaporation, which allowed for the approximation of the real values of the size of the evaporation surface. In the work presented, this was the way in which the near-surface rate of copper was measured, and how its surface was measured. Additionally, in the existing literature, there are no experimental results of the discussed copper refinement process conducted using CCF technology.

Table 1 shows the base results of current experiments regarding the removal of lead from liquid copper during refinement conducted inside vacuum induction furnaces.

## 3. Materials and Methods

### 3.1. Materials Used in the Research

Examinations were conducted on copper alloys, the compositions of which are shown in Table 2.

### 3.2. Research Apparatus

The research programme was realized using two vacuum induction furnaces manufactured by SECO-WARWICK (Świebodzin, Poland): namely, a vacuum induction crucible furnace ICF (with a ceramic crucible) and an induction furnace with a cold crucible CCF. The baseline technical and construction parameters of both devices are shown in Table 3.

### 3.3. Research Methodology and Parameters

A sample of the examined alloy with a mass of 3300 g was placed inside the crucible. Depending on the type of the furnace, it was either an Isopress 1126-MG951 crucible produced by Capital Refractories made out of MgO, or a cold crucible cooled with water made from copper. The inside diameter of both crucibles was equal to 90 mm. Each of the experiments described in this work was conducted following the same approach. In the first stage, after inserting the sample into the furnace, the pressure was lowered inside the chamber and then heating was turned on and the insert was melted. After achieving the desired temperature of liquid metal, it was kept inside the furnace for 600 s. The temperature was measured using a thermoelectric sensor, type PtRh30-PtRh6, and an optical pyrometer. During the process, samples of liquid metal were extracted at the defined time intervals. After the experiment was finished, the chemical composition of the metal was examined. For this purpose, atomic absorption spectrometry was used, using an ASA solar device.

In the ICF aggregate, the process was analyzed for the temperature range of 1323–1473 K; for the CCF aggregate, the temperature range was 1273–1323 K. The furnace’s working power was in the range of 30–40 kW for the ICF, and it was equal to 170 kW for the CCF. The working pressure for both aggregates varied within the range of 10 to 1000 Pa.

The overall coefficient of lead mass transport (*k*_Pb_) in the analyzed process can be described using Equation (1):(1)$$\frac{1}{k_{Pb\;\;}}=\frac{1}{\beta^{l\;\;}}+\frac{1}{k_{e\;}}+\;\frac{1}{\beta^{g\;}}$$
where:
*β^l^*—the mass transfer coefficient of Pb in liquid phase, m·s^−1^,*β^g^*—the mass transfer coefficient of Pb in gas phase, m·s^−1^,*k_e_*—the evaporation rate coefficient, m·s^−1^.

Assuming that the process of ingredient evaporation from a liquid metallic alloy can be generally described by a first-degree kinetic equation, the value of the *k_Pb_* was calculated using Equation (2):(2)$$\frac{dC_{Pb}}{dt}=k_{Pb}\cdot \frac{F}{V}\cdot C_{Pb}$$
which, after transformation, can be shown as (3):(3)$$2.303\log\frac{C_{Pb}^{t}}{C_{Pb}^{0}}=-k_{Pb}\frac{F}{V}\left(t-t_{0}\right)$$
where:
$C_{Pb}^{0}\;i\;C_{Pb}^{t}$—the lead concentration in copper: initial and at time t, wt. %,*F*—the evaporation surface, m^2^,*V*—the metal volume, m^3^,(*t* − *t*_0_)—the duration of the process, s.

In the case of metal that is melted and mixed inside an induction furnace, to calculate the coefficient value for the transport of lead inside the liquid phase *β^l^*, the Machline equation [23] is recommended. This equation takes the form of (4):(4)$$\beta^{l}={\left(\frac{8\;D_{Pb\;}\vartheta_{m}}{\pi \;r_{m}}\;\right)}^{\;0,5}$$
where:
*ν*_m_—the surface velocity of the inductively mixed liquid metal, m s^−1^,*r*_m_—the radius of the surface of the liquid metal (most often taken as the inner radius of the crucible), m,*D*_Pb_—the lead diffusion coefficient in liquid copper, m^2^ s^−1^,

Analyzing the evaporation rate of the load from the liquid copper surface, it was assumed that the maximum value of the rate constant *k*^e^_Pb_ can be described by Equation (5):(5)$$k_{Pb}^{e}=\frac{\alpha \cdot \;p_{Pb\;}^{0}\;\gamma_{Pb}\cdot \;M_{Cu}}{{\left(2\pi RTM_{Pb}\right)}^{0,5}\;\cdot \;\rho_{Cu}}$$
where:
*α*—the evaporation constant,*p*^0^_Pb_—the equilibrium vapor pressure of lead over the liquid alloy, Pa,*γ*_Pb_—the activity factor of lead in liquid copper,*M*_Cu_, *M*_Pb_—the molar masses, respectively, of copper and lead, g mol^−1^,*ρ*_Cu_—the density of copper, g m^−3^.

To estimate the resistance values for the process of mass transport in the liquid phase (*R*^l^) and the resistance related to the reaction happening on the liquid alloy surface (*R*^e^) in the overall evaporation process resistance, Equations (6) and (7) were used:(6)$$R^{l}=\frac{\left(\frac{1}{\beta^{l}}\right)}{\left(\frac{1}{k_{Pb}}\right)}\;\;\cdot \;\;100\%$$
(7)$$R^{e}=\frac{\left(\frac{1}{k^{e}}\right)}{\left(\frac{1}{k_{Pb}}\right)}\;\;\cdot \;\;100\%\;$$

## 4. Results and Discussion

Table 4 and Table 5 show the results of the experimental smelting of the examined alloys. These tables show the final lead concentration values in the alloy (*C*^k^_Pb_), the percentage mass loss of lead (*m*), and the calculated values of the evaporation stream of lead (*N*). A graphical interpretation of these results is shown in Figure 1, Figure 2 and Figure 3.

Analyzing the results shown in Table 3 and Table 4 and in Figure 1, Figure 2 and Figure 3 it, can be concluded that, for the alloy with a higher lead concentration, a higher level of this metal is removed from copper. This is due to the higher pressure balance of lead above the alloy with a high concentration of this metal. The maximum relative mass loss of lead from the alloy was observed for the alloy containing 8% of Pb per mass; it appeared within the full range of the used smelting parameters (temperature, pressure) within the range of 87% to 97% for the crucible furnace and 87–94% for the cold crucible aggregate. Comparatively, the relative lead loss from the alloy containing 1.9% of Pb by mass was between 69% to 86% for the crucible furnace and 55–79% for the cold crucible aggregate. For the experiments conducted inside a cold crucible furnace, close values of lead removal were observed in the lower temperatures (1273–1323 K). This means that the analyzed process takes place faster within this aggregate.

Additionally, the study results show an increase in the effectiveness of the lead removal process as the pressure was lowered within the smelting aggregate. At the same time, there was a strong difference in the density of the stream of removed lead. The approximated values for this parameter for samples for the crucible furnace were between 6.14 × 10^−4^ g/(cm^2^ s) and 3.54 × 10^−4^ g/(cm^2^ s), and for the cold crucible the samples were between 2.23 × 10^−4^ g/(cm^2^ s) and 1.67 × 10^−4^ g/(cm^2^ s). This was caused by the significant difference in the size of the molten metals in both aggregates.

To measure the stages that determine the rate of the examined evaporation process, kinetic analysis was conducted, based on its experimental results.

From a kinetic point of view, the process of lead evaporation from liquid copper can be divided into three stages:Lead transport from inside liquid copper to the surface;The phenomenon of evaporation from the surface of liquid copper;The transport of lead vapor into the gas phase.

From Equation (3) it can be concluded that, to calculate the value of the coefficient *k*_Pb_, knowledge of both the size of the bath and its surface area are required. In the experiments analyzed, in the temperature range of 1273–1473 K, the estimated value of the total liquid metal volume was in the 1.21 to 1.26 × 10^−4^ m^3^ range. The density values of liquid copper in the analyzed temperature range were calculated based on the relations contained within the work [24]. A harder task was estimating the real evaporation surface (the surface of the liquid copper). In the case of a metal or alloy melted inside an induction furnace (ICF or CCF Technology, Swiebodzin, Poland), the size of the metal’s surface is dependent on the way the electromagnetic field interacts with the metallic bath, as well as the properties of the liquid metal [25,26,27,28,29]. This is showcased by the creation of a prominent meniscus on the surface of the bath. To visualize this, Figure 4 shows the surface of liquid copper inside an induction crucible furnace.

To estimate the real surface size of the liquid metal, the method described in the paper [30,31] was used. It consists of four stages:Creating photos of molten metal using a fast camera;Establishing the geometry of the created meniscus based on these photos;Charting a function graph that describes the meniscus during different parameters of the process;Estimating the surface area using Wolfram Mathematica and MicroStation software.

Estimations of metal surface area based on the described method are variable within the range of 113 to 131 cm^2^ for samples inside a crucible furnace, and between 227 and 230 cm^2^ for experiments inside a cold crucible furnace. For comparison, the inside surface area of the crucible used measured at 63.6 cm^2^.

Calculated from Equation (3), the values of the overall lead transport coefficient k_Pb_ are shown in Table 6 and Table 7. Figure 5 presents an example graphical interpretation of the change in the *k*_Pb_ coefficient as a function of the working pressure of an induction crucible furnace.

From the data shown in Table 6 and Table 7, it can be concluded that, for all of the experiments undertaken, the estimated values of the overall mass transport coefficient *k*_Pb_ were within the range of 1.66 × 10^−5^ to 5.80 × 10^−5^ ms^−1^. These results concern both the experiments conducted inside an inductive crucible furnace and those conducted inside a cold crucible furnace. Reducing the working pressure of the melting aggregates and the temperature caused the increase in the value of this coefficient, which means a higher rate of lead evaporation. The *k*_Pb_ coefficient values achieved during the study are in line with the ones in the table.

### 4.1. Mass Transfer in the Liquid Phase

Machlin assumed, based on the analogy of penetrative mass insertion models, that elements of liquid metal move according to a tangent: liquid metal–gas or liquid metal–crucible, whereby the gradient of the rate normal to the surface is close to zero. From this correlation, it can be concluded that the value of the *β*^l^ is directly proportional to the value of the surface rate of the liquid metal. Most authors researching the evaporation kinetics of inductively mixed metal bath ingredients assume, as Machlin did, that this rate is not related to the electric parameters of the furnace. For induction furnaces with volumes up to one ton, it was assumed that the *ν_m_* is constant and measures at 0.1 ms^−1^. However, later works determined that this rate is related to parameters such as furnace power, current frequency, crucible geometry, or the position of the crucible inside the furnace compared to the inductor [32,33]. In the work presented here, to calculate the value of the *β*^l^ coefficient, rate values were taken from [34], which were estimated for different metal alloys in experiments conducted in the same kind of melting aggregate as used in the experiment described above. For liquid copper ϑ*_m_*, the rate values were between 0.114 and 0.135 ms^−1^. In Table 6 and Table 7, the *β*^l^ coefficient values are shown, and were calculated using Equation (4). They fell within the range of 1.40–1.78 × 10^−4^ ms^−1^. The lead diffusion coefficients from liquid copper were estimated using [35,36].

### 4.2. The Process of Evaporation from the Interface

The values of lead vapor pressure *p*^0^_Pb_ presented in this work were calculated based on thermodynamics [37]; data from [38] were used to calculate the value of lead activity inside liquid copper *γ_Pb_*. Based on these estimates, for the temperature range of 1273–1473 K, the values of *p*^0^_Pb_ ranged between 0.91 × 10^2^ and 1.91 × 10^3^ Pa, and the activity coefficient values ranged from 4.52 to 7.25. The *k*^e^_Pb_ coefficient values calculated from Equation (5) are shown in Table 3 and Table 4.

Figure 6 and Figure 7 show the change in the influence of resistances *R*^l^ and *R*^e^ in the overall resistance of the evaporation process, depending on the working pressure inside the melting aggregate.

Based on the data shown in Figure 6 and Figure 7, it can be concluded that, for the analyzed temperature ranges and for pressure above 10 Pa, the lead evaporation rate is practically limited by mass transport in the gas phase. The overall contribution of resistance from mass insertion into the metallic liquid phase in the overall process resistance does not exceed 33%. As the pressure lowers, the contribution of this resistance increases. This happens in the experiments conducted in both smelting aggregates. From the data shown in these Figures, it can be concluded that, for all experiments, the influence of *R*^e^ resistance on the overall process resistance is between 9 and 28%

For tests conducted inside the crucible furnace, the overall resistance contribution of transport resistance related to the reaction happening on the surface of liquid copper *R*^e^ and resistance of mass transport inside the liquid phase *R*^l^ in the overall resistance of the evaporation process of lead were within the range of 21 to 61%. The summarized values of resistance *R*^l^ + *R*^e^ exceeded 50% for tests conducted at a pressure of 10 Pa.

For tests conducted inside a furnace with a cold crucible, the summarized mass transport resistance in the liquid phase *R*^l^ in the overall resistance of the lead evaporation process did not exceed 34%. This means that, in this case, the lead evaporation rate is determined mainly by mass transport in the gaseous phase.

## 5. Conclusions

The presented results allow us to formulate the following conclusions:The process of refining copper from lead conducted in vacuum induction furnaces happens faster inside an aggregate with a cold crucible, as compared to a typical crucible furnace. This is caused by a significant difference in the size of the metal bath’s surface area. The calculated surface areas of the meniscus formed during smelting alloys using a furnace with a cold crucible are significantly larger compared to those where the inserts were melted in induction crucible furnaces.For the experiments conducted inside an induction crucible furnace, lowering the working pressure inside the furnace chamber from 1000 to 10 Pa while increasing the temperature from 1323 to 1473 K was accompanied by a drop in the lead concentration inside the alloy of between 69 and 96%, compared to its initial mass. At the same time, the value of the overall mass transport coefficient *k_Pb_* increased from 3.50 × 10^−5^ to 5.27 × 10^−5^ ms^−1^.For the experiments conducted inside a cold crucible furnace, approximate values of lead removal appeared for lower temperatures (1273 to 1323 K), confirming that the analyzed process happens faster in this aggregate.When the pressure is lowered, the rate-determining step of the analyzed process changes. When the pressure is equal to 10 Pa, the value of resistance related to mass transport in the liquid phase (*R^l^*) and the reaction on the surface of the liquid metal (*R^e^*) comprises nearly 60% of the overall process resistance.For the experiments conducted inside a cold crucible surface, in the overall temperatures and pressures used, the rate of the process is limited by mass transport in the gas phase.

## Figures and Tables

**Figure 1 materials-15-07024-f001:**
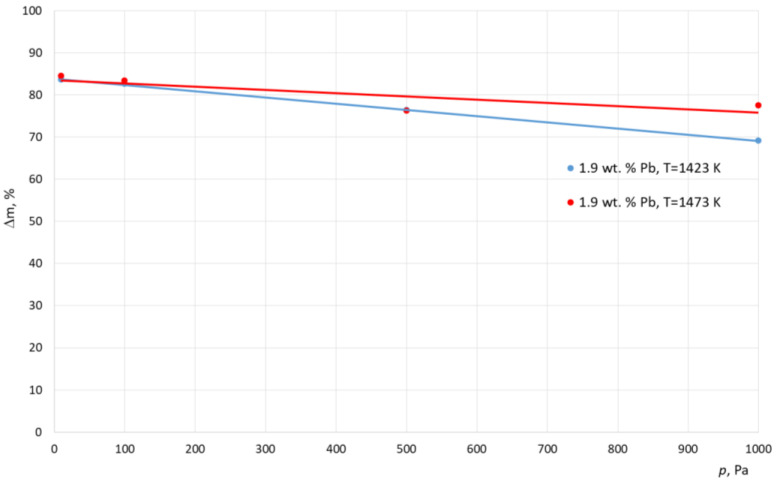
Relative mass loss of lead during smelting realized inside ICF, with different working pressures (alloy 1.9 wt. % Pb).

**Figure 2 materials-15-07024-f002:**
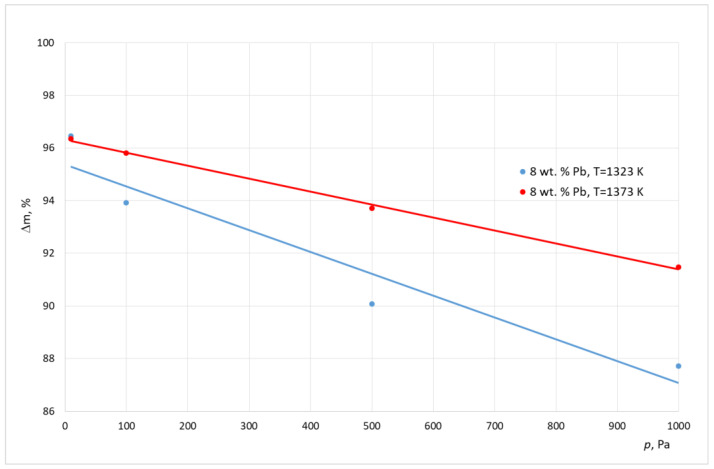
Relative mass loss of lead during smelting realized inside ICF, with different working pressures (alloy 8 wt. % Pb).

**Figure 3 materials-15-07024-f003:**
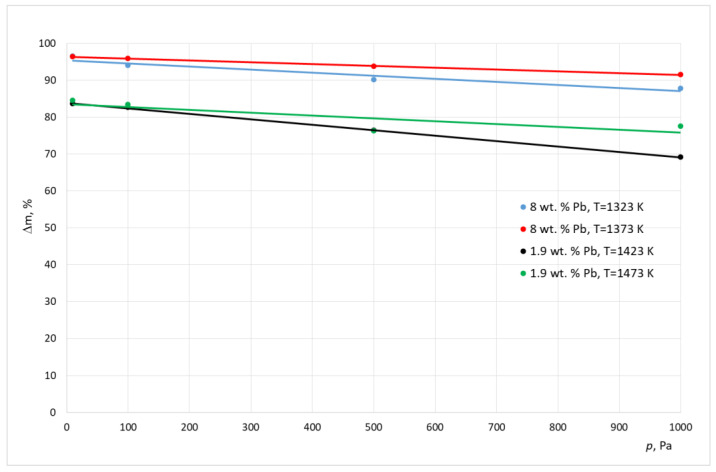
Relative mass loss of lead during smelting realized inside CCF, with different working pressures.

**Figure 4 materials-15-07024-f004:**
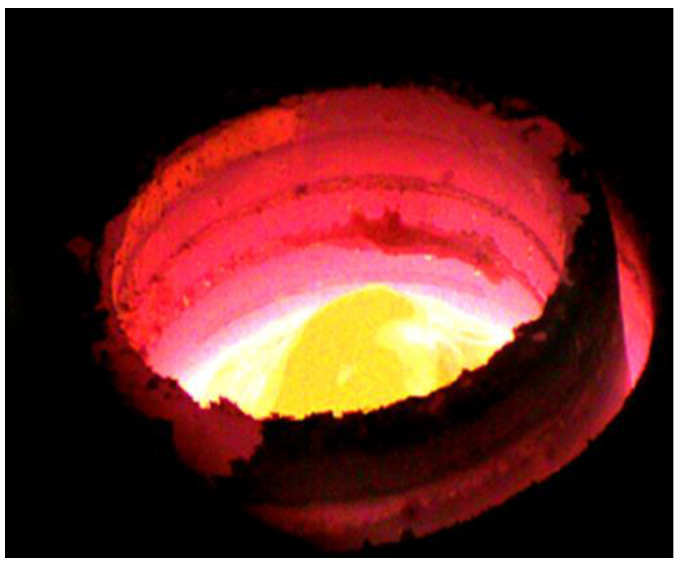
Photo of the surface of molten copper inside an induction furnace. The furnace’s working power is 53 kW [30].

**Figure 5 materials-15-07024-f005:**
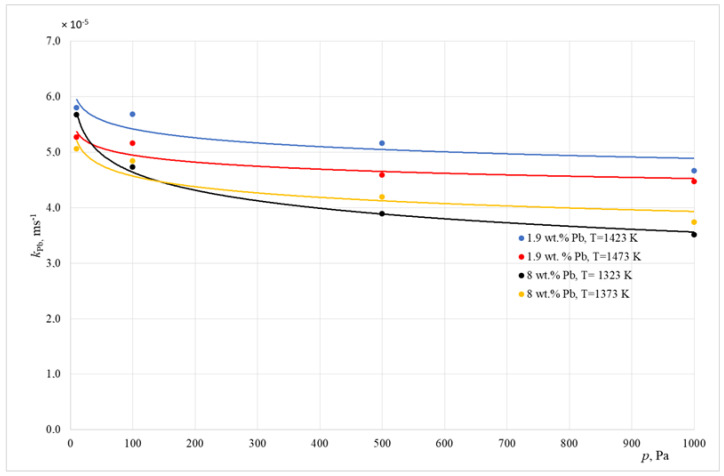
Change in the *k*_Pb_ value as a function of the induction melting unit pressure (ICF).

**Figure 6 materials-15-07024-f006:**
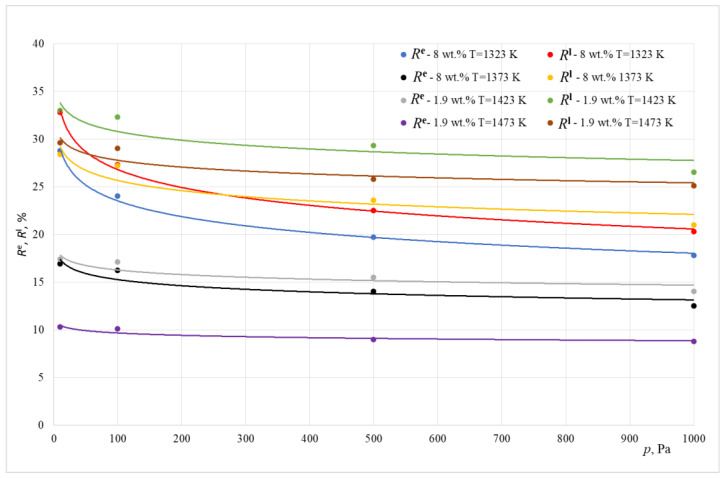
The share of the *R^e^* and *R^l^* resistance in the overall resistance of the evaporation process (ICF).

**Figure 7 materials-15-07024-f007:**
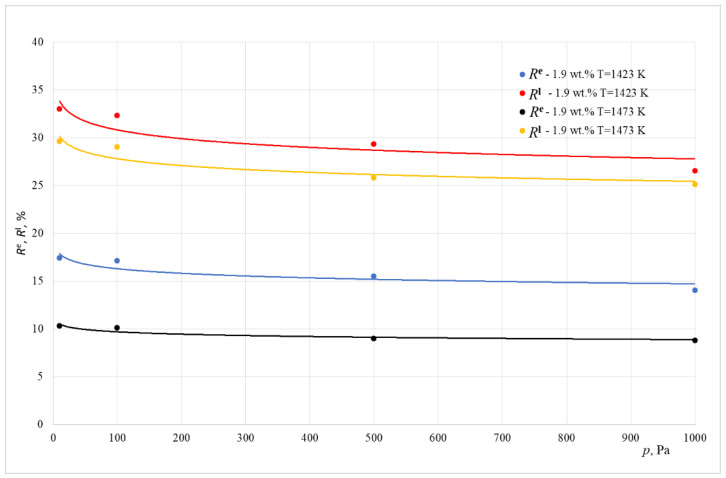
The share of the *R^e^* and *R^l^* resistance in the overall resistance of the evaporation process (CCF).

**Table 1 materials-15-07024-t001:** Base parameters and results of examinations of the process of the removal of copper through refinement in vacuum induction furnaces.

	Literature No.	Material	Pb Content, wt.%	Temperature Range, K	Pressure Range, Pa	Pb Removal Rate, %	Mass Transport Coefficient, *k*_Pb_·10^5^, m·s^−1^
1	[15]	Cathode copper enriched with Sb and Pb	<0.01	1373÷1423	13÷26.000	<90	-
2	[16]	Cathode copper	<0.0015	1573	13	>90	-
		Cathode copper enriched with Sb and Pb	<0.01	1503÷1523	0.13÷13	>90	-
3	[17,18]	Blister copper, anode and cathode copper	0.027 ÷ 0.33	1423÷1523	8÷40	<95	1.5÷4.5
4	[19,20,21]	Synthetic Cu-Pb alloys, blister copper	<2	1373÷1523	8–1333	<90	0.6–8.2
5	[22]	Synthetic Cu-Pb alloys	<2	1473÷1623	10.000–55.000	<48	1.09–2.11

**Table 2 materials-15-07024-t002:** Chemical composition of the alloys used in this study.

Alloy Designation	Alloy Component, wt. %.								
	Pb	Sn	Al	Fe	Mn	Ni	Si	*p*	Cu
CuPb8	8	10	<0.02	<0.2	<0.2	<0.2	<0.02	<0.05	residue
CuPb1.9	1.9	12	<0.01	<0.2	<0.2	<2	<0.01	<0.4	residue

**Table 3 materials-15-07024-t003:** Technical and construction parameters of the aggregates used in the present study.

Device Parameters	ICF (VIM)	CCF (ISM 2-200)
Maximum power	75 kW	200 kW
Maximum vacuum	0.01 Pa	0.001 Pa
Vacuum system	Mechanical pump, roots pump, diffusion pump	Mechanical pump, roots pump, diffusion pump
Maximum working temperature	2073 K	2073 K
Crucible volume	2.5 dm^3^	1 dm^3^

**Table 4 materials-15-07024-t004:** Results of experiments carried out in an induction vacuum furnace.

Alloy with an Initial Lead Content	T, K	*P*, kW	*p*, Pa	*C*^k^_Pb_ wt. %	*m*, %	*N*·10^4^, g/cm^2^·s
Cu-Pb 1.9 wt.% Pb	1423	30	1000	0.59	69.12	6.2
			500	0.46	76.40	7.0
			100	0.34	82.55	7.8
			10	0.32	83.59	7.8
	1473	40	1000	0.49	77.47	6.1
			500	0.46	76.20	6.4
			100	0.32	83.33	7.1
			10	0.30	84.54	7.1
Cu-Pb 8 wt.% Pb	1323	30	1000	1.06	87.71	3.24
			500	0.86	90.07	3.40
			100	0.53	93.91	3.54
			10	0.31	96.45	3.53
	1373	40	1000	0.74	91.46	3.21
			500	0.55	93.70	3.23
			100	0.36	95.80	3.29
			10	0.32	96.35	3.32

**Table 5 materials-15-07024-t005:** Results of experiments carried out in an induction furnace with a cold crucible.

Alloy with an Initial Lead Content	*T*, K	*P*, kW	*p*, Pa	*C*^k^_Pb_ wt. %	*m*, %	*N*·10^4^, g/cm^2^·s
Cu-Pb 1.9 wt.% Pb	1323	170	1000	0.84	55.78	2.23
			500	0.66	65.26	2.41
			100	0.54	71.57	2.48
			10	0.39	79.47	3.17
Cu-Pb 8 wt.% Pb	1273	170	1000	0.99	87.62	1.61
			500	0.77	90.37	1.63
			100	0.62	92.12	1.64
			10	0.42	94.75	1.67

**Table 6 materials-15-07024-t006:** The overall mass transfer coefficient *k*_Pb_ and coefficient *β*^l^_Pb_ and *k*^e^_Pb_ (ICF).

Alloy with an Initial Lead Content	T, K	*p*, Pa	*k_Pb_*·10^5^, m·s^−1^	*β*^l^_Pb_·10^4^, m·s^−1^	*k*^e^_Pb_·10^4^, m·s^−1^
Cu-Pb 1.9 wt.% Pb	1423	1000	4.66	1.76	3.33
		500	5.15	1.76	3.33
		100	5.67	1.76	3.33
		10	5.80	1.76	3.33
	1473	1000	4.47	1.78	5.11
		500	4.58	1.78	5.11
		100	5.16	1.78	5.11
		10	5.27	1.78	5.11
Cu-Pb 8 wt.% Pb	1323	1000	3.50	1.73	1.97
		500	3.89	1.73	1.97
		100	4.73	1.73	1.97
		10	5.66	1.73	1.97
	1373	1000	3.74	1.78	2.99
		500	4.19	1.78	2.99
		100	4.84	1.78	2.99
		10	5.05	1.78	2.99

**Table 7 materials-15-07024-t007:** The overall mass transfer coefficient *k*_Pb_ and coefficient *β*^l^_Pb_ and *k*^e^_Pb_ (CCF).

Alloy with an Initial Lead Content	T, K	*p*, Pa	*k_Pb_*·10^5^, m·s^−1^	*β*^l^_Pb_·10^4^, m·s^−1^	*k*^e^_Pb_·10^4^, m·s^−1^
Cu-Pb 1.9 wt.% Pb	1323	1000	1.73	1.40	1.97
		500	1.82	1.40	1.97
		100	1.86	1.40	1.97
		10	2.12	1.40	1.97
Cu-Pb 8 wt.% Pb	1273	1000	1.66	1.38	1.02
		500	1.67	1.38	1.02
		100	1.69	1.38	1.02
		10	1.94	1.38	1.02

## Data Availability

Data sharing is not applicable to this article.

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
