# Peer review of "Comparative Analysis of Lead Removal from Liquid Copper by ICF and CCF Refining Technologies"

_materials, 2022, doi:10.3390/ma15197024_

Round 1

Reviewer 1 Report

Q1: The data processing method in line 208-210 is interesting. It is suggested that the obtained data be visualized ( mapped ) and reflected in figure 4, which makes the analysis of line 213-216 convincing.

Q2: The figures and tables in the manuscript have repeated meanings. It is recommended to retain one of the figures or tables.

Author Response

Dear Reviewer,

On behalf of my co-authors, we are very grateful to you for giving us an opportunity to revise our manuscript. we appreciate you very much for your positive and constructive comments and suggestions on our manuscript entitled “Comparative Analysis of Lead removal from liquid copper by ICF and CCF refining technologies”.

We have studied your comments carefully and tried our best to revise our manuscript according to the comments. Below are our responses and revisions to your questions and suggestions point by point. Thank you again for your hard work !

Reviewer 2 Report

Reviewer’s Report

This work aims at comparing the effectiveness of two metal refining methodologies, ICF and CCF. This is a very interesting topic with high scientific and technological relevance. However, some modifications are needed before accepting this work for publication. Suggested modifications are as follow:

Title

The title should better read “Comparative Analysis of Lead removal from liquid copper by ICF and CCF refining technologies”

Abstract

Results obtained in the experiment should be captured in the abstract. Recommendations too must be captured in the abstract.

Introduction

1. The introduction should contain the gap in knowledge of the research or significance of the research.

2. Correct the definition of ICF in line 41.

State of art

1. Ln 49 – 50, “Presented in these the results were limited only to data showing the change of concentration of the removed additives during the smelting process” This sentence should be rephrased.

2. Ln 73 “The results of copper vacuum refinement, especially mentioning lead are the papers” Correct this sentence

3. Ln 78 -80 “As the pressure lowers ... the liquid phase increased” Recast this sentence

4. Ln 81 - 84 “Analyzing literature data ... speed of liquid copper stirred using induction” This sentence is too long, ambiguous and confusing. Please, recast them into two sentences for better understanding.

Experimental

1. Always use upper case “T” and “F” for Tables and Figures in research reporting.

Research Methodology

1. Ln 116 - 118 “The inside diameter was equal for both crucible and measured at 90 mm. Realizing each of the experiments described in the article was conducted using a constant schematic” Should be rephrased to be more understandable.

2. Ln 122 “During the process, samples of liquid metal were extracted in measured amounts of time” Rephrase this sentence. The meaning is not clear.

3. Ln 127 - 129 “The registered change of furnace working power for ICF measured between 30 to 40 kW, for CCF measured 170 kW” should be recast to make more meaning. Presently, it is ambiguous.

Results and discussion

1. Percentage mass loss of lead was defined as (m) but in Table 4, it was represented with ∆m, are the two symbols the same? Correct it with the right one and be consistent.

2. There are two “Ps” in Table 4, why did you not define and differentiate them. It is confusing if left so.

3. Why are you duplicating results? If the information contained in Table 4 is the same as that contained in Figure 1,  then I don’t know why the duplication. Either you present it as a Table or you present it as a Figure and discuss what is contained;  so also in Figures 2 and 3.

4. Ln 149 – 167: Please, the discussion of Figure 1-3 and Tables 4 and 5 should be repeated in more understandable English. I mean it should be rephrased for clarity. As it is, I don’t understand what the authors are trying to portray. When results are being discussed, observations encountered in an experiment should be explained. So that in a bid to explain such observations, similar trends or opposing trends from other researchers should be cited. It is only through citation of other works that your work will be scientifically sound and acceptable. Research is not done in isolation. 

5. I suggest that equations 1 – 7 and their definitions should be presented in methodology section. Only their results should appear under this section.

6. Ln 280 - 281 “Presented in the work values of lead vapor pressure p0Pb data was calculated with used the thermodynamic base” recast this sentence

7. Ln 300 – 316. Can the authors discuss this observation more technically by giving reasons for all that were observed and citing one or two authors who have observed such. This discussion is just like a higher college student who is reporting chemistry practical. Make it more scientific and appealing to prospective readers. Then again, improve on the English.   

Conclusion

Please, redo conclusion and make it clearer and more readable.

References

Your references are too obsolete. The most recent citation was one work published in 2020, followed by one in 2018 and 2017. All others are 10 years older than the present work. Improve on the references.

Author Response

Dear Reviewer,

On behalf of my co-authors, we are very grateful to you for giving us an opportunity to revise our manuscript. we appreciate you very much for your positive and constructive comments and suggestions on our manuscript entitled “Comparative Analysis of Lead removal from liquid copper by ICF and CCF refining technologies”.

We have studied your comments carefully and tried our best to revise our manuscript according to the comments. Below are our responses and revisions to your questions and suggestions point by point. Thank you again for your hard work !

Response to reviewer comments

Thank you for your kind comments and suggested modifications:

  1. Title

Response 1: According to the suggestion the title has been changed.

  1. Abstract Results obtained in the experiment should be captured in the abstract. Recommendations too must be captured in the abstract.

Response 2: The abstract was extended accordingly to the suggestion.

Introduction

  1. The introduction should contain the gap in knowledge of the research or significance of the

research.

Response 3: The itroduction was extended accordingly to the suggestion

  1. Correct the definition of ICF in line 41.

Response 4: The definition of ICF was modified. The correction has been made in the revised text.

State of art

  1. Ln 49 – 50, “Presented in these the results were limited only to data showing the change of concentration of the removed additives during the smelting process” This sentence should be rephrased.
  2. Ln 73 “The results of copper vacuum refinement, especially mentioning lead are the papers” Correct this sentence
  3. Ln 78 -80 “As the pressure lowers ... the liquid phase increased” Recast this sentence
  4. Ln 81 - 84 “Analyzing literature data ... speed of liquid copper stirred using induction”

This sentence is too long, ambiguous and confusing. Please, recast them into two sentences

for better understanding.

Response 5: The correction has been made in the revised text.

Experimental

  1. Always use upper case “T” and “F” for Tables and Figures in research reporting.

Response 6: The correction has been made in the revised text.

Research Methodology

  1. Ln 116 - 118 “The inside diameter was equal for both crucible and measured at 90 mm.

Realizing each of the experiments described in the article was conducted using a constant

schematic” Should be rephrased to be more understandable.

  1. Ln 122 “During the process, samples of liquid metal were extracted in measured amounts

of time” Rephrase this sentence. The meaning is not clear.

  1. Ln 127 - 129 “The registered change of furnace working power for ICF measured between

30 to 40 kW, for CCF measured 170 kW” should be recast to make more meaning. Presently,

it is ambiguous.

Response 7: The correction has been made in the revised text.

Results and discussion

  1. Percentage mass loss of lead was defined as (m) but in Table 4, it was represented with

∆m, are the two symbols the same? Correct it with the right one and be consistent.

Response 8: ∆m was defined mass loss m was defined as mass.

  1. There are two “Ps” in Table 4, why did you not define and differentiate them. It is

confusing if left so.

Response 9: P (upper) is power, p (lower) is pressure.

  1. Why are you duplicating results? If the information contained in Table 4 is the same as that contained in Figure 1, then I don’t know why the duplication. Either you present it as a Table or you present it as a Figure and discuss what is contained; so also in Figures 2 and 3.

Response 10: Thank you for your comments. The table contains additional data than those presented in the chart. Graphical presentation of the results enables the reader to precisely see the trend of changes in the measured quantity.

  1. Ln 149 – 167: Please, the discussion of Figure 1-3 and Tables 4 and 5 should be repeated in more understandable English. I mean it should be rephrased for clarity. As it is, I don’t understand what the authors are trying to portray. When results are being discussed, observations encountered in an experiment should be explained. So that in a bid to explain such observations, similar trends or opposing trends from other researchers should be cited. It is only through citation of other works that your work will be scientifically sound and acceptable. Research is not done in isolation.

Response 11: The correction has been made in the revised text.

  1. I suggest that equations 1 – 7 and their definitions should be presented in methodology

section. Only their results should appear under this section.

Response 12: The correction has been made in the revised text.

  1. Ln 280 - 281 “Presented in the work values of lead vapor pressure p0Pb data was

calculated with used the thermodynamic base” recast this sentence

Response 13: The correction has been made in the revised text.

  1. Ln 300 – 316. Can the authors discuss this observation more technically by giving reasons

for all that were observed and citing one or two authors who have observed such. This

discussion is just like a higher college student who is reporting chemistry practical. Make it

more scientific and appealing to prospective readers. Then again, improve on the English.

Response 14: The correction has been made in the revised text.

Conclusion

Please, redo conclusion and make it clearer and more readable.

Response 15: Conclusions are reconstructed accordingly to suggestions.

References

Your references are too obsolete. The most recent citation was one work published in 2020,

followed by one in 2018 and 2017. All others are 10 years older than the present work.

Improve on the references.

 Response 16: According to the suggestion references has been changed.

We would like to thank you again for taking the time to review our manuscript. Your suggestions are of great guiding significance for our paper writing and scientific research !

Sincerely,

Albert Smalcerz

Reviewer 3 Report

The work presented by the authors is good. But they show very little interest to format their paper. In many places, the instructions given by the editorial office is noted in the manuscript. This shows the quality of checking by the authors.

1. Author Contributions: For research articles with several authors. a short paragraph specifying their individual contributions must be provided. The following statements should be used “Conceptualization. L.B. and A.S.; methodology. L.B.; software. D.D.; validation. J.L. B.W. and M.J.; formal analysis. A.S.; investigation. B.W.; resources. B.W.; data curation. D.D.; writing—original draft preparation. L.B.; writing—review and editing. A.S. and J.L.; visualization. D.D and M.J. All authors haveread and agreed to the published version of the manuscript.”

2. References
1. Author 1. A.B.; Author 2. C.D. Title of the article. Abbreviated Journal Name Year. Volume. page range.
2. Author 1. A.; Author 2. B. Title of the chapter. In Book Title. 2nd ed.; Editor 1. A.. Editor 2. B.. Eds.; Publisher: Publisher Location. Country. 2007; Volume 3. pp. 154–196. 358
3. Author 1. A.; Author 2. B. Book Title. 3rd ed.; Publisher: Publisher Location. Country. 2008; pp. 154–196.
4. Author 1. A.B.; Author 2. C. Title of Unpublished Work. Abbreviated Journal Name year. phrase indicating stage of publication (submitted; accepted; in press).
5. Author 1. A.B. (University. City. State. Country); Author 2. C. (Institute. City. State. Country). Personal communication. 2012.
6. Author 1. A.B.; Author 2. C.D.; Author 3. E.F. Title of Presentation. In Proceedings of the Name of the Conference. Location of Conference. Country. Date of Conference (Day Month Year).
7. Author 1. A.B. Title of Thesis. Level of Thesis. Degree-Granting University. Location of University. Date of Completion.
8. Title of Site. Available online: URL (accessed on Day Month Year).

3. Reference section is Too weak. Out of 23 articles reviewed, 17 are very old. So the readers can not get better knowledge from these old references.

4. Avoid the usage of abbreviations in title.

5. What is ICF2?

6. ICF and CCF are very old and known methodologies. Where is the novelty?

Author Response

Dear Reviewer,

On behalf of my co-authors, we are very grateful to you for giving us an opportunity to revise our manuscript. we appreciate you very much for your positive and constructive comments and suggestions on our manuscript entitled “Comparative Analysis of Lead removal from liquid copper by ICF and CCF refining technologies”.

We have studied your comments carefully and tried our best to revise our manuscript according to the comments. Below are our responses and revisions to your questions and suggestions point by point. Thank you again for your hard work !

Response to reviewer comments

The work presented by the authors is good. But they show very little interest to format their paper. In many places, the instructions given by the editorial office is noted in the manuscript. This shows the quality of checking by the authors.

  1. Author Contributions: For research articles with several authors. a short paragraph specifying their individual contributions must be provided. The following statements should be used “Conceptualization. L.B. and A.S.; methodology. L.B.; software. D.D.; validation. J.L. B.W. and M.J.; formal analysis. A.S.; investigation. B.W.; resources. B.W.; data curation. D.D.; writing—original draft preparation. L.B.; writing—review and editing. A.S. and J.L.; visualization. D.D and M.J. All authors haveread and agreed to the published version of the manuscript.”

 Response 1: The correction has been made in the revised text.

  1. References
    1. Author 1. A.B.; Author 2. C.D. Title of the article. Abbreviated Journal Name Year. Volume. page range.
    2. Author 1. A.; Author 2. B. Title of the chapter. In Book Title. 2nd ed.; Editor 1. A.. Editor 2. B.. Eds.; Publisher: Publisher Location. Country. 2007; Volume 3. pp. 154–196. 358
    3. Author 1. A.; Author 2. B. Book Title. 3rd ed.; Publisher: Publisher Location. Country. 2008; pp. 154–196.
    4. Author 1. A.B.; Author 2. C. Title of Unpublished Work. Abbreviated Journal Name year. phrase indicating stage of publication (submitted; accepted; in press).
    5. Author 1. A.B. (University. City. State. Country); Author 2. C. (Institute. City. State. Country). Personal communication. 2012.
    6. Author 1. A.B.; Author 2. C.D.; Author 3. E.F. Title of Presentation. In Proceedings of the Name of the Conference. Location of Conference. Country. Date of Conference (Day Month Year).
    7. Author 1. A.B. Title of Thesis. Level of Thesis. Degree-Granting University. Location of University. Date of Completion.
    8. Title of Site. Available online: URL (accessed on Day Month Year).

            Response 2: The correction has been made in the revised text.

  1. Reference section is Too weak. Out of 23 articles reviewed, 17 are very old. So the readers can not get better knowledge from these old references.

 Response 3: Reference was added.

  1. Avoid the usage of abbreviations in title.

Response 4: Inserting the full names of the furnaces instead of abbreviations would significantly lengthen the title of the article.

  1. What is ICF2?

Response 5: I did not find such a phrase in the text.

  1. ICF and CCF are very old and known methodologies. Where is the novelty?

Response 6: I agree with the reviewer, both technologies have been known for a long time. A novelty is the use of them, primarily CCF, in refining processes. Hence, there are few publications on their use.

We would like to thank you again for taking the time to review our manuscript. Your suggestions are of great guiding significance for our paper writing and scientific research !

Sincerely,

Albert Smalcerz

Round 2

Reviewer 2 Report

I recommend its acceptance

Reviewer 3 Report

All the queries are addressed by the authors.